# Autologous Blood Doping Induced Changes in Red Blood Cell Rheologic Parameters, RBC Age Distribution, and Performance

**DOI:** 10.3390/biology11050647

**Published:** 2022-04-23

**Authors:** Marijke Grau, Emily Zollmann, Janina Bros, Benedikt Seeger, Thomas Dietz, Javier Antonio Noriega Ureña, Andreas Grolle, Jonas Zacher, Hannah L. Notbohm, Garnet Suck, Wilhelm Bloch, Moritz Schumann

**Affiliations:** 1Department of Molecular and Cellular Sports Medicine, Institute of Cardiovascular Research and Sports Medicine, German Sport University Cologne, Am Sportpark Müngersdorf 6, 50933 Cologne, Germany; emily.zollmann@freenet.de (E.Z.); janina.bros@t-online.de (J.B.); benediktseeger@web.de (B.S.); thomas-fritz.dietz@gmx.de (T.D.); h.notbohm@dshs-koeln.de (H.L.N.); w.bloch@dshs-koeln.de (W.B.); m.schumann@dshs-koeln.de (M.S.); 2German Red Cross Blood Donation Service West, Center for Transfusion Medicine Hagen, Feithstraße 184, 58097 Hagen, Germany; j.noriega@bsdwest.de (J.A.N.U.); a.grolle@bsdwest.de (A.G.); garnetsk@gmail.com (G.S.); 3Department of Preventive and Rehabilitative Sports and Performance Medicine, Institute of Cardiovascular Research and Sports Medicine, German Sport University Cologne, Am Sportpark Müngersdorf 6, 50933 Cologne, Germany; j.zacher@dshs-koeln.de

**Keywords:** autologous blood doping, red blood cells, red blood cell rheologic parameters, exercise performance, red blood cell age

## Abstract

**Simple Summary:**

Autologous blood doping (ABD) refers to the artificial increase in circulating red blood cell (RBC) mass by sampling, storage, and transfusion of one’s own blood. It is assumed that some athletes apply this prohibited technique to improve oxygen transport capacity and thus exercise performance. The primary aim of this study was to test whether RBC rheological and associated parameters significantly change due to ABD with the consideration of whether this type of measurement might be suitable for detecting ABD. Further, it was assessed whether those changes are translated into indices of endurance performance. Eight males underwent an ABD protocol combined with several blood parameter measurements and two exercise tests (pre and post transfusion). Results of this investigation suggest a change in the distribution of age-related RBC sub-populations and altered deformability of total RBC as well as of the respective sub-populations. Further, the identified changes in RBC also appear to improve sports performance. In conclusion, these data demonstrate significant changes in hematological and hemorheological parameters, which could be of interest in the context of new methods for ABD detection. However, additional research is needed with larger and more diverse study groups to widen the knowledge gained by this study.

**Abstract:**

Autologous blood doping (ABD) refers to the transfusion of one’s own blood after it has been stored. Although its application is prohibited in sports, it is assumed that ABD is applied by a variety of athletes because of its benefits on exercise performance and the fact that it is not detectable so far. Therefore, this study aims at identifying changes in hematological and hemorheological parameters during the whole course of ABD procedure and to relate those changes to exercise performance. Eight healthy men conducted a 31-week ABD protocol including two blood donations and the transfusion of their own stored RBC volume corresponding to 7.7% of total blood volume. Longitudinal blood and rheological parameter measurements and analyses of RBC membrane proteins and electrolyte levels were performed. Thereby, responses of RBC sub-populations—young to old RBC—were detected. Finally, exercise tests were carried out before and after transfusion. Results indicate a higher percentage of young RBC, altered RBC deformability and electrolyte concentration due to ABD. In contrast, RBC membrane proteins remained unaffected. Running economy improved after blood transfusion. Thus, close analysis of RBC variables related to ABD detection seems feasible but should be verified in further more-detailed studies.

## 1. Introduction

Autologous blood doping (ABD) refers to the transfusion of one’s own blood after it has been stored. The World Anti-Doping Agency (WADA) enlisted ABD as M1. Manipulation of blood and blood components, which is prohibited at all times. Blood doping was banned by the International Olympic Committee (IOC) in 1985, although detection methods did not exist at that time [1]. Still, ABD has been used by professional athletes since the 1960s to improve their performance [2] by increasing and optimizing oxygen delivery to the muscles, due to an increased circulating red blood cell (RBC) mass and thus, elevated arterial oxygen content [3,4,5,6]. This was related to reduced lactate concentration during submaximal exercise intervention and increased maximal oxygen uptake (VO_2_max) (see for review [5]). The ABD procedure includes donation of larger blood volumes several weeks prior to competition to enable the recovery of the initial RBC mass. The RBC are preferably cryopreserved and stored at −80 °C, as studies revealed less reduction in cell quality compared to conventional storage conditions at refrigerator level [7,8,9,10,11]. The transfusion of the stored RBC back to the donor/athlete usually takes place 1–7 days before the competition [12]. 

Although ABD has been used by athletes for decades, testing strategies or distinct markers evidencing this procedure in vivo are still under investigation. Currently, applied attempts to detect ABD include the athlete’s biological passport (ABP), which, among others, consists of a longitudinal monitoring of blood parameters [13,14]. However, the ABP requires ongoing development because the hematological parameters including hemoglobin concentration (Hb) or hematocrit (hct) might not only change in response to ABD but also due to environmental factors such as altitude training [15,16]. Other developments [17] including the measurement of micoRNA fingerprints [18], proteomics to detect possible changes in RBC membrane protein content [19], measurements of plasticizers from the blood bags [20] or detecting markers of blood age [21] or RBC extracellular vesicles [22] seem promising; however, they might require further development in order to doubtlessly prove ABD. 

Investigations of RBC rheologic parameters indicated prominent changes of RBC deformability and RBC viscosity of cryopreserved RBC after blood donation, and after in vitro simulation of transfusion (mixture of stored and fresh RBC), thus suggesting that these variables are most likely affected during ABD in general [7,8,9]. In addition, mean cellular volume (MCV) was shown to increase during cryopreservation, resulting in unfavorable surface-to-volume ratio; and also electrolytes such as sodium, potassium and calcium, which also determine RBC volume, were affected by cryopreservation [8]. However, while previous studies were able to indicate altered RBC deformability after blood donation in vivo and of cryopreserved RBC, they did not show altered RBC rheological parameters after in vivo transfusion of cryopreserved RBC. It was thus concluded that the transfusion volume was too low (<4% of total blood volume) for the modified cryopreserved RBC to have a major impact on the RBC rheological parameters of the entire circulating RBC pool [7]. 

Although promising, evidence of changes in RBC rheological parameters after transfusion of cryopreserved RBC is less pronounced thus far. Thus, the primary aim of the present investigation was to test whether RBC rheological and associated parameters are significantly affected during blood donations and transfusion of cryopreserved RBC if the transfusion volume is increased compared to previous studies. In addition, detection methods not yet applied in this context, such as measurements of RBC deformability under an osmotic gradient and analysis of RBC according to cell age, were applied. As a secondary aim, it was assessed whether changes in RBC rheology are translated into indices of endurance performance, such as maximal oxygen consumption, metabolic thresholds and exercise economy. 

## 2. Materials and Methods

### 2.1. Study Group

The study participants were recruited via announcements in social networks or on the homepage of the German Sport University Cologne. The following exclusion criteria were defined: smoking and use of regular medications. Alcohol consumption was prohibited at least 48 h prior to blood sampling/donation and exercise tests. Further, diseases including diabetes mellitus type I/II, and blood disorders such as hemophilia or hemoglobinopathies were excluded as well as blood donation within the last twelve (12) weeks because of possible interferences with the study parameters. Intake of (dietary) supplements that might affect the study parameters (e.g., iron, EPO, Vitamin A/B/C, copper) was also prohibited. The risks and benefits of the study were explained to the participants, after which the subjects gave their written and oral consent. The study was approved by the ethics committee of the German Sport University Cologne and aligned with the Declaration of Helsinki (#016/2019).

After the selection process, eight healthy men were enrolled in the study. The anthropometric data were as follows: age 20.5 ± 3.7 years, weight 80.1 ± 11.2 kg and height 1.82 ± 0.07 m. Mean VO_2_max was 51.0 ± 7.7 mL/min/kg, and mean training hours per week were reported to be 9.5 ± 7.3 h. One participant used protein shakes once per week, and one reported the use of creatine and Omega-3.

### 2.2. Overview of Study Design: ABD Procedure, Blood Sampling and Exercise Testing

Blood donation, cryopreservation, storage and RBC transfusion took place at the German Red Cross Blood Donation Service, Hagen, Germany under standardized and quality controlled conditions by experienced physicians. Blood samplings for the measurement of study parameters were carried out at the German Sport University Cologne by medical staff.

To determine baseline values, 20 mL of venous blood was sampled into sodium heparin vacutainer (BD Vacutainer^®^, 17IU/mL blood; Beckton Dickinson GmbH, Heidelberg, Germany) for immediate analyses of the study parameters described below (=T0).

One week after these baseline measurements, one blood unit (=500 mL) was sampled from each subject into conventional blood bags containing the common anticoagulation solution CPDA-1 (citrate, phosphate, dextrose, adenine). The blood bags were automatically processed to separate the RBC, which were subjected to cryopreservation and storage at −80 °C using the automated cell processor ACP^®^215 (Haemonetics^®^, Munich, Germany). Thus, Glicerolo (Haemonetics^®^, Munich, Germany) was added as freezing medium representing a high-glycerol method (40% glycerol proportion in the RBC units) in order to protect the RBC during freezing, storage and thawing. 

Fifteen (15) weeks after T0, another 20 mL of venous blood was sampled into sodium heparin vacutainer for the direct measurements of the study parameters (=T1).

One week later, a second blood unit was sampled from each participant, processed, cryopreserved and stored as described above. 

After another 12 weeks, study parameters were measured after sampling 20 mL of venous blood (=T2) into sodium heparin vacutainer. Additionally, exercise testing was conducted (Exercise test_Pre; for description, see below) approximately 1 h after blood sampling. Two weeks later, both stored RBC units were thawed and washed by the automated cell processor ACP^®^215 (Haemonetics^®^, Munich, Germany) and transfused back to the respective donor; thus, donor and recipient were identical. The participants were allowed to drink and eat between RBC transfusion and the following blood collection time point. Two hours (=T3), 24 h (=T4), three days (=T5) and seven days (=T6) after RBC transfusion, 20 mL blood was sampled from the participants into sodium heparin vacutainer for immediate measurements of the study parameters, respectively. At T6, the exercise test was repeated (Exercise Test_Post) one hour after blood sampling (Figure 1). 

### 2.3. Complete Blood Count

A complete blood count was measured from anticoagulated whole blood using the automatic cell analyzer Sysmex Digitana KX-21N (Sysmex, Horgen, Switzerland) and the following RBC parameters are presented: hemoglobin concentration (g/L), hematocrit (%) and RBC distribution width (RDW) (%).

### 2.4. Calculation of RBC Volume

Direct measurements of RBC volume require invasive methods including radioactive isotopes, which was not applicable in this study. Thus, RBC volume was estimated by calculation using the subjects’ hematocrit, height, weight and gender. Thus, the Nadler equation for men was applied: blood volume = (0.3669 × H^3^) + (0.03219 × W) + 0.6041 [23].

### 2.5. RBC Deformability and Osmoscan

RBC deformability was measured at various fluid shear stresses by laser diffraction analysis using the laser assisted optical rotational red cell analyzer (Lorrca MaxSis, RR Mechatronics, Zwaag, The Netherlands). The automated Lorrca allows the measurement of both temperature-controlled RBC deformability measurements by ektacytometry and osmoscan measurements, which refer to measurements of RBC deformability under an osmotic gradient [24,25]. 

For RBC deformability measurements, a standardized amount of 100 mio RBC were mixed with 5 mL of an isotonic viscous medium (0.14 mM Polyvinylpyrrolidone, PVP, 29 cP at 37 °C, RR Mechatronics, Zwaag, The Netherlands). For this purpose, the required volume was calculated from the RBC count obtained during the measurement of the complete blood count. The mixed samples were sheared in a Couette system composed of a glass cup and a precisely fitting bob, with a gap of 0.3 mm between the cylinders. A laser beam was directed through the sheared sample, and the diffraction pattern produced by the deformed cells was automatically analyzed and transformed into an Elongation Index (EI). Nine consecutive shear stresses between 0.3–30 Pa were applied to the samples resulting in nine EI values presented by the software. Finally, the Lorrca software 5.04 calculated the maximum deformability at infinite shear stress (EImax) by the use of the nine EI. Thus, higher EImax values reflect higher RBC deformability.

For the osmoscan measurements, a standardized amount of one billion RBC were mixed with 5 mL of an isotonic PVP medium (see above). The RBC deformability under an osmotic gradient (=Osmoscan) was automatically measured using the Lorrca MaxSis. Results included Omin, reflecting the osmolality at minimum RBC deformability, beyond which RBC would lyse with a further decrease in osmolality. EImax represents the maximum deformability at isotonicity, and Ohyper reflects the hyperosmotic osmolality corresponding to 50% EImax.

### 2.6. Immunohistochemical Staining of Membrane Proteins

A published immunohistochemical protocol [26] was used to investigate the levels of the RBC membrane proteins Na^+^/K^+^-ATPase, Glycophorin A and Aquaporine 3. Briefly, RBC in whole blood were fixed with 4% formaldehyde, separated by centrifugation (120× *g*, 3 min, RT), dispersed on a slide and heat fixed. Each slide contained a control and a test area. Slides were washed using tris buffered saline (TBS), and the RBC membrane was permeabilized for 30 min at 37 °C using 0.1% trypsin (except for Glycophorin A staining). Exogenous peroxidase was applied to block unspecific binding sites prior to incubation of the samples with the primary antibodies. The test areas were incubated with either Rabbit anti sodium potassium ATPase (1:100; abcam, Berlin, Germany), Mouse anti Glycophorin A (1:500; Sigma-Aldrich/Merck, Taufkirchen, Germany) or Rabbit anti Aquaporine 3 (1:200; abcam, Berlin, Germany) for 1 h at RT. The control areas were incubated without the respective primary antibody. Slides were washed using TBS and both the control, and the test area was incubated with the respective secondary antibody (Na^+^/K^+^-ATPase and Aquaporine 3: goat anti-rabbit antibody; 1:150; Dako, Glostrup, Denmark; Glycophorin A: goat anti-mouse antibody; 1:150; Dako, Glostrup, Denkmark). The staining was developed using 3,3-diaminobenzidine-tetrahydrochloride solution (Sigma-Aldrich, St. Louis, MO, USA), dehydrated in increasing alcohol solutions, mounted using Entellan^®^ (Merck, Darmstadt, Germany) and covered. Images of the stained RBC were taken using an Axiophot 1 microscope (Zeiss, Oberkochen, Germany) coupled to a camera (Progres Gryphax Prokyon; Jenoptik Optical Systems GmbH, Jena, Germany) with a 400-fold magnification. Stainings were analyzed with the software ImageJ 1.52a (National Institutes of Health, Bethesda, MD, USA). For staining intensity analysis, mean grey values were measured, and total immunostaining intensity was calculated as previously described [27].

### 2.7. RBC Viscosity and Aggregation 

An aliquot of whole blood was separated by centrifugation (3700× *g* for 10 min at 4 °C), white blood cells and platelets were removed and RBC were resuspended in 1% BSA (Merck Chemicals GmbH, Darmstadt, Germany) in 0.1 mol PBS (pH 7.4) to receive a final hematocrit of 40%. 

Sample viscosity was determined using the Brookfield cone-plate viscometer DV2T (AMETEK GmbH/B.U. Brookfield, Hadamar-Steinbach, Germany). The sensed resistance to the rotation of the cone produces a torque that is proportional to the shear stress in the fluid. This measure was converted by the system to absolute centipoise units (cP = mPa·s). The viscosity of the sample was measured at a shear rate of 150 1^−s^. 

For aggregation measurement, samples were fully oxygenated prior to measurement. Aggregation index (%) was then measured at 37 °C by syllectometry using the Lorrca MaxSis (RR Mechatronics, Zwaag, The Netherlands). Oxygenated samples were transferred to the Couette system, and changes of backscattered light were recorded over 120 sec and presented as a graph to calculate an aggregation index (AI%) [28]. 

### 2.8. RBC Electrolyte Concentrations

Another aliquot of whole blood was separated by centrifugation (3700× *g* for 10 min at 4 °C), and RBC were diluted (1:10) with phosphate buffered saline (PBS; 0.1 mol; pH 7.4). RBC Na^+^, K^+^ and Ca^2+^ concentrations were measured using flame photometry (EFOX 5053; Eppendorf AG, Hamburg, Germany).

### 2.9. Glycerol Levels

A third aliquot of whole blood was separated as described above. RBC glycerol concentration was measured at T0 and T3 only, after 1:100 dilution of RBC with PBS (0.1 mol; pH 7.4) using the Glycerol GK Assay Kit according to the manufacturers’ instructions (Megazyme/Neogen Europe Ltd., Auchincruive, Scotland).

### 2.10. Density Gradient Centrifugation

To separate the whole RBC population into young RBC (light, highly flexible), old RBC (dense, less flexible) and the main fraction (remaining RBC population), whole blood was subjected to percoll (VWR, Darmstadt, Germany) density gradient centrifugation as previously described [29]. Briefly, percoll solutions containing 56%, 52%, 48%, 46% of percoll stock solution (density 1.130 g/mL) and remaining SAH-buffer (26% BSA, 13.2 mM NaCl, 4.6 mM KCl and 10 mM HEPES) were layered on top of each other starting with the densest solution. Whole blood was separated by centrifugation (3500× *g* for 5 min at room temperature (RT)) and RBC were washed using GASP-buffer (9mM Na_2_HPO_4_, 1.3 mM NaH_2_ PO_4_*H_2_O, 140mM NaCl, 5.5 mM glucose and 0.8% BSA) and finally diluted 1:2 with SAH buffer. Then, 600 μL of the prepared blood suspension was cautiously transferred on top of the percoll gradient and centrifuged at 3700× *g* for 30 min at 4 °C. The RBC layers consisted of different RBC subpopulations with young RBC within the top layers, old RBC in the bottom layer and main fraction in the layers in between. The layers were transferred to clean tubes, washed 1:2 with GASP buffer and centrifuged (800× *g*, 10 min, 4 °C). The supernatant was discarded, and the RBC pellets were weighed to calculate the percent distribution of young RBC, main fraction and old RBC. The RBC were then used for RBC deformability and osmoscan measurements as described above.

### 2.11. Exercise Testing

Exercise testing on a treadmill was carried out at T2 and T6. The testing session consisted of a submaximal incremental test, followed by a maximal ramp test. After a 5 min warm-up at 1.5 m/s, the incremental test started at 2 m/s with a 1% incline. Speed was increased every three minutes by 0.5 m/s. After each increment, the treadmill was stopped for 45 s to allow for capillary blood sampling. The submaximal incremental test was terminated after participants reached a blood lactate concentration of 4 mmol/L. Capillary blood samples during the incremental test were collected from the earlobe into hemolyzing solution cups (EKF Diagnostic Sales, Magdeburg, Germany). Blood lactate concentration was measured using the EKF Biosen C-Line Analyser (EKF Diagnostics GmbH, Barleben, Germany). Following 5 min of active recovery at walking pace, a maximal ramp test was performed. Starting velocity was individually determined at a speed of 0.4 m/s below the increment in which participants reached 4 mmol/L in the incremental test. Speed increased every minute by 0.2 m/s, and participants were verbally encouraged to give maximal performance. The test was terminated at voluntary exhaustion. Maximal exhaustion was defined by common criteria reported elsewhere [30]. During both tests, cardiorespiratory data were recorded using a stationary breath by breath gas analyzer (Metalyzer^®^ 3B; Cortex Biophysik GmbH, Leipzig, Germany). Data were interpolated to receive values for every second. VO_2_max was defined as the highest 30 s moving average oxygen uptake during the ramp test. Running speed at 4 mmol/L was determined from the incremental test by interpolation Additionally, to measure running economy, oxygen uptake was averaged for the last minute of each increment of the incremental test. 

### 2.12. Statistical Analysis

The software Graph PadPrism 8.0.2 (GraphPad Software, San Diego, CA, USA) was used for statistical analyses and preparation of the graphs. Presented data are mean ± standard deviation. Normal distribution of the data was tested using the Shapiro–Wilk test because of its high statistical power when working with small sample sizes [31]. Repeated measures ANOVA with Tukey’s multi-comparison test was applied to detect possible differences in the tested parameters during blood sampling and after transfusion. Ordinary one-way ANOVA was performed for the osmoscan data of old RBC because of missing values. An overall type I error rate of 0.05 was used as an indication of statistical significance for each calculation and the following labeling was chosen */^#^
*p* < 0.05; **/^##^
*p* < 0.01; ***/^###^
*p* < 0.001. Additionally, Cohens d effect size was calculated for type I error rates between 0.051–0.07. Comparisons of RBC volume between T2 and T3, comparison of glycerol concentration between T0 and T3 and performance parameters between T2 and T6 were tested using paired *t* test. Correlation analyses were performed to test for a relation of the proportion of young RBC on RBC deformability of all RBC and between RBC volume and maximum oxygen uptake.

## 3. Results

### 3.1. Transfusion Volume 

Both cryopreserved RBC units resulted in an average of 413.1 ± 34.1 mL RBC that were transfused to the respective recipient (=donor). According to the estimated blood volume, this transfusion volume corresponded to 7.7 ± 1.2% of total blood volume. 

### 3.2. RBC Parameters

Hemoglobin concentration and hct significantly increased after blood transfusion (T3). Values increased by +10.0 ± 6.3 g/dL and +6.4 ± 5.2% compared to T2, and by +7.5 ± 5.5 g/dL and +5.4 ± 6.1% compared to T0. Values remained at this higher level until the end of the investigation period (T6). However, at T6, one week after transfusion, values tended to decrease. The values remained unaltered from T0 to T1 and T2, suggesting that the blood volumes were restored between the two donations, respectively (Figure 2A). RDW was significantly affected by the intervention. Values at T3 were higher compared to T0 (ES = 0.5; moderate effect). Values at T6 were significantly lower compared to T3 (*p* < 0.05) and T4 (*p* < 0.05), respectively (Figure 2B). Estimated RBC volume significantly increased after transfusion in all participants (*p* < 0.001) (Figure 2C). Distribution of the RBC subpopulations young RBC, main fraction and old RBC is presented in Figure 2D. Percentage of young RBC significantly increased by +84% at T1 (*p* < 0.001 vs. T0), and values remained high during the study period. Percentage of main fraction RBC significantly decreased by −34% from T0 to T1 (*p* < 0.01), and levels were pervasively lower throughout the study. Amount of old RBC significantly decreased by −54% from T0 to T1 (*p* < 0.001), and reduced values were observed until T3. Then, percentage of old RBC increased again after RBC transfusion (T4–T6). Overall, transfusion (T3) resulted in a −3.2% decrease in young RBC, a +4.6% increase in main fraction RBC and a −1.3% decrease in old RBC. The changes were not significantly different compared to T2.

### 3.3. RBC Rheologic Parameters Aggregation, Viscosity, and Deformability of the Total RBC Population and RBC Deformability Measures of RBC Subpopulations

The aggregation index was not affected by the intervention (Figure 3A). RBC viscosity at hct 40% significantly decreased from T0 to T1 (*p* < 0.05), and the values remained at this lower level during the study (Figure 3B). Maximum deformability (EImax) of total RBC continuously increased during the course of the study. Distinct increases were observed from T0 to T2 (ES = 0.79, high effect) and from T0 to T6 (*p* < 0.05) (Figure 3C). EImax of young RBC significantly increased from T1 to T2 (*p* < 0.05), decreased at T3 (*p* < 0.05) and then again increased until the end of the study (*p* < 0.05 T6 vs. T3) (Figure 3D). EImax of the main RBC fraction showed a similar course as young RBC. However, increased values at T2 and the drop at T3 were not statistically significant. Values significantly increased from T3 to T5 (*p* < 0.05) and T6 (*p* < 0.05), respectively (Figure 3E). EImax of old RBC was significantly decreased at T3 (*p* < 0.05) and remained reduced until T6 (Figure 3F). Correlation analysis revealed a positive relation of the proportion of young RBC with EImax of total RBC (* *p* < 0.05), suggesting that a younger RBC population increases the overall RBC deformability.

### 3.4. RBC Osmoscan of Total RBC and RBC Subpopulations

RBC osmoscan of total RBC revealed higher Omin at T3 (*p* < 0.01 vs. T1) and higher values at T6 compared to T0 (*p* < 0.05) and T1 (*p* < 0.05), respectively (Figure 4A). In contrast, Omin decreased in young RBC until T5 after which the values increased again (*p* < 0.001 vs. T4; *p* < 0.01 vs. T3) (Figure 4B). Omin of the main RBC fraction showed less changes, but values at T6 were significantly higher compared to T0 (*p* < 0.05; Figure 4C). Omin of old RBC significantly increased from T0 to T2 (*p* < 0.05). Thereafter, the values decreased. Not all samples of old RBC could be measured, not because they were missing, but because the quality of the old RBC was apparently too poor for the system to record the data (Figure 4D). EImax (osmoscan) of total RBC was not altered during the study period (Figure 4E), while values of young RBC increased from T0 to T3 (*p* < 0.05) (Figure 4F). In addition, EImax (osmoscan) of the main RBC fraction significantly increased from T0 to T4 (*p* < 0.05) (Figure 4G). EImax (osmoscan) of old RBC slightly decreased until T2 but data were not significantly different. Again, some data are missing because of the poor quality of old cells (Figure 4H). Ohyper shows small but significant reductions in total RBC. Values at T1 (*p* < 0.01) and T4 (*p* < 0.05) were lower compared to T0, respectively (Figure 4I). Similar observations were made for young RBC. Here, Ohyper was significantly reduced at T3 (*p* < 0.05) and T4 (*p* < 0.05) compared to T0, respectively. Ohyper of the main fraction was significantly lower at T3 (*p* < 0.05), T4 (*p* < 0.05) and T5 (*p* < 0.01) when compared to T0. Values increased from T4 to T6 (*p* < 0.01) (Figure 4K). Reduction of Ohyper of old RBC was not statistically significant (Figure 4L). 

### 3.5. Immunostaining of RBC Membrane Proteins, Glycerol and Electrolyte Levels within RBC

Signal of immunostaining for the membrane proteins Glycophorin-A and Aquaporin-3 remained unaltered during the investigation period (Figure 5A,B). RBC glycerol concentration significantly increased from T0 to T3 (*p* < 0.01) (Figure 5C). Staining for Na^+^/K^+^-ATPase was unaffected by the intervention (Figure 6A). RBC sodium concentration first decreased (*p* < 0.001 T0 vs. T1), then increased (*p* < 0.01 T2 vs. T1) and then decreased again (*p* < 0.05 T3 vs. T2) (Figure 6B). RBC potassium level increased from T0 to T3 (*p* < 0.05) and then decreased until T5 (*p* < 0.01) (Figure 6C). RBC calcium concentration continuously decreased during intervention (Figure 6D).

### 3.6. Exercise Performance 

Blood lactate concentrations were significantly reduced at all velocities at T6 compared to T2 (Figure 7A). Speed at 4 mmol/L significantly increased by 0.19 ± 0.13 m/s (+5.17 ± 3.77%) from T2 to T6 (*p* < 0.01). Furthermore, oxygen uptake was lower across all increments at T6 compared to T2 (Figure 7B). At T6, oxygen consumption at a running speed of 2.0 m/s and (−9.7 ± 12.4%, *p* < 0.05) and 2.5 m/s was significantly reduced (−7.7 ± 7.2%, *p* < 0.05) compared to T2. Maximal running speed was significantly higher at T6 compared to T2 (+3.0 ± 2.8%, *p* < 0.05). Relative VO_2_max did not statistically differ between T2 (51.0 ± 7.7 mL/min/kg) and T6 (51.9 ± 7.0 mL/min/kg; *p* > 0.05; ES: 0.64). However, correlation analyses indicated a positive relation between VO_2_max and RBC volume (*p* < 0.05).

## 4. Discussion

ABD is used by elite athletes because of its benefits in terms of endurance performance [4,5,6]. A study by Faiss and colleagues reported a blood doping prevalence of approximately 15–18% in the investigated track and field athletes [32]. Similar values of abnormal hematological profiles were published by Stray-Gundersen et al. [33] for cross-country skiers. The authors further indicated an even higher prevalence of highly abnormal hematological values among medal winners, while the values were most likely normal among those athletes who only finishes below place 41 [33]. According to these data, blood doping might be highly prominent among those athletes competing for the place on the podium. Although it is known that athletes use ABD to gain an advantage in competition, there are still no valid ABD detection methods and thus, new approaches are needed to complement the existing strategies. The study examined RBC rheological parameters and associated variables during each phase of the ABD procedure, thus providing a longitudinal observation to test whether ABD distinctively affects the parameters. 

The key findings of the present investigation indicate prominent changes of the investigated hematological parameters during the transfusion phase and thereafter. Further, to the best of our knowledge, these are the first data showing a shift in the proportion of RBC sub-populations—young RBC, old RBC and main fraction—toward a higher percentage of young RBC after the donation of large blood volumes. From the tested RBC rheological parameters, RBC deformability was the only parameter distinctively affected by the ABD protocol. ABD altered RBC deformability of RBC in general, but also of the respective sub-populations. Electrolyte levels of potassium and calcium were highly altered during the intervention, while in contrast, RBC membrane proteins that are thought to affect RBC deformability in a certain way were not altered during ABD. In addition, running speed at 4 mmol/L as well as maximal running speed increased, and running economy improved after blood transfusion.

Improvements in exercise performance after ABD using cryopreserved RBC is well described and [6] persists up to four weeks after blood transfusion [4]. Similar results were observed herein. The conducted submaximal test revealed reduced blood lactate concentrations during the increments one week after transfusion compared to pre-transfusion level thus, supporting recent findings [5]. Further, sub-maximal oxygen consumption was lower after transfusion, suggesting improved exercise-related economy. In contrast to previous studies, the recent data did not support findings of increased maximum oxygen uptake after transfusion [5]. However, while not statistically significant, the effect size of 0.64 suggests a small increase in VO_2_max. Moreover, correlation analysis revealed a significantly positive relation between VO_2_max and RBC volume, which directly relates improved performance to RBC transfusion. 

ABD-dependent improved performance was accompanied by distinct increases in hematological variables. However, the literature data indicate that the measured blood values were still within the normal range of individual variations and legal limits [4], which might explain the difficulty in detecting ABD by hematological analyses. Measurements of hematological parameters, including hemoglobin concentration and hematocrit, within the present study showed comparable values between baseline and 14 (T1) respective 12 (T2) weeks after blood donations. Sampling of large blood volumes immediately reduces hematocrit and hemoglobin concentration. This decrease is augmented by repeated blood donations [34]. However, increased erythropoietin release after blood sampling [35] stimulates erythropoiesis and allows the restoration of initial RBC mass approximately 9 weeks after blood donation [36]. Exact calculations of blood and RBC volume require the application of radioactive isotopes [23], which were not suitable in the recent study. Calculation of the RBC and blood volume using the Nadler equation might be less accurate compared to the “Gold Standard”, but it is widely applied [23] and thus also used herein. The results indicated that blood and RBC volumes were restored within the scheduled recovery period as suggested in order to exploit additive effects of transfused RBC. Upon transfusion (T3), hematocrit increased by 6.42 ± 5.2% (range −1.6 to + 15.2%). One participant showed a hematocrit value of 52.2%, thus exceeding the cut-off level set to 50% by several sports federations [37]. Similar results were obtained from hemoglobin concentration measurements. Values significantly increased at T3 by 10.0 ± 6.3 g/dL (range 2.7–19.9 g/dL), and the maximum value was found to be 17.5 g/dL; thus, none of the values exceeded the cut-off level of 18.5 g/dL [38]. This increased niveau persisted until the end of the study. Only continuous monitoring of individual values might detect blood manipulation but might be highly difficult in practical application. In addition, high fluid intake might mask blood manipulation, and diurnal variations of blood parameters [39] must also be taken into account. Thus, additional data are needed to complement hematological measurements. Data analyses of RBC distribution width, a marker for RBC volume and size distribution, revealed only small variations with none of the values exceeding the normal range (11.8–14.5%). RDW increased during the blood donation phase, indicating a wider distribution of RBC volume/size; possibly because of the increased erythropoiesis described earlier, since the mean cellular volume of young RBC was reported to be higher than of old RBC [29]. RDW decreased after transfusion, and it might be speculated whether transfusion of the stored RBC volume might be sufficient to induce this change. Effects of blood donation on distribution of RBC sub-populations were measured via percoll density gradient centrifugation. This technique separates the RBC according to their density and not by their actual age. That’s why the sub-populations were labeled as “young” (less dense), “main fraction” and “old” (very dense) [29]. The remarkable increase in circulating young RBC after the first blood donation, and the enduring higher levels further indicate stimulated erythropoiesis [36]. Surprisingly, a further increase in circulating young RBC was not observed several weeks after the second blood donation, which might require a more detailed investigation right after blood donation. The proportion of old RBC and main fraction decreased at T1 accordingly. The proportion of old RBC slightly dropped by 1.3% from T2 to T3, while the proportion of main RBC fraction increased by 4.6%. This might suggest that the age of transfused RBC was shifted toward a younger RBC population as reported earlier [40]. 

Endurance training has also been shown to shift RBC age distribution towards a younger RBC population [41,42,43,44,45], and thus, it should be considered that top athletes show a training-induced increase in circulating young RBC per se. However, when applying comparable determination methods, the reported percentage of young RBC in endurance athletes (~20%) [42] was lower than that reported herein after the first blood donation (~55%), suggesting that blood donation might represent a higher erythropoiesis stimulus compared to endurance training. A younger RBC population might be of advantage because of a superior surface-to-volume ratio and higher RBC deformability compared to senescent cells [29]. These characteristics seem to improve the cell entry into small capillaries and the cell ability to traverse small capillaries [46]. RBC deformability of RBC in total increased during the blood donation phase from T0 to T2, which was correlated with the higher proportion of circulating young RBC. Increased deformability was also observed in young RBC and RBC main fraction. Transfusion induced only a temporary drop in RBC deformability and only in young RBC because values increased again one day after transfusion (T4). The opposite development was observed in old RBC. Further information on RBC deformability changes during ABD was obtained by RBC deformability measurements under an osmotic gradient. EImax at isotonicity increased in young RBC and main fraction RBC, while smaller effects were observed for old RBC or total RBC. Omin of total RBC increased during the study, which corresponds to a right shift in osmolality at which 50% of the cells lyse in a hypo-osmotic environment. This increase might be related to the 100% increase in Omin observed in old RBC between T0 and T2, which might not be compensated by the value reduction of young RBC, since this reduction was <5%. Still, values of total RBC remained within the range previously published [47]. Ohyper showed a graduation between young RBC, main fraction and old RBC with lowest values observed in old RBC, reflecting recent data [47]. Values reduced during the intervention, which might be related to changes in intracellular water [47]. Overall, the data suggest a positive effect of blood donation on RBC deformability because it leads to a higher proportion of young RBC, which was not affected by transfusion. 

As stated above and by others, hemodynamic parameters are affected by blood donation and transfusion of high blood volumes [48]. Thus, an increase in hematocrit might also affect blood viscosity [49] and RBC aggregation parameters [50]. More precisely, a relation between elevated hematocrit and hyperviscosity as well as reduced heart function was described for transfused blood volumes of more than 8% of total blood volume [51]. These values were not achieved in this study with transfused RBC volumes < 8% of total blood volume. Data of the present study confirm that RBC aggregation and viscosity were not affected by the intervention, and thus, hyper-viscosity and hyper-aggregation were less likely. This was also in accordance to findings by Çinar et al. [52], who report that the hematocrit must increase by more than 10% to increase viscosity by 20%, thus significantly reducing blood flow rate to pose a reasonable health risk. 

To further explain changes in RBC deformability during the study, RBC membrane proteins were investigated. The RBC membrane protein glycophorin A is the major transmembrane sialoglycoprotein, and its ligation was shown to negatively affect RBC deformability [53]. For conventional blood storage, it was shown that glycophorin A levels decrease during storage [54], while such investigations have not been conducted in cryopreserved cells. Data of the present investigation indicate that glyocophorin A is not significantly affected by blood donation or transfusion of cryopreserved RBC. However, a small increase in staining intensity was observed from T0 to T1. Given the fact that glycophorin A levels are higher in young RBC than in old RBC [55], this small increase might be related to the higher percentage of young RBC reported after the first blood donation. Aquaporin-3 was measured because it was indicated that this integral membrane protein might be responsible for the glycerol movement across the RBC membrane [56]. Since glycerol was used as cryoprotectant and because a previous study indicated that possibly higher RBC glycerol levels might also account for altered RBC rheologic properties observed during cryopreservation [8], it was thought that cryopreservation might either change aquaporine-3 content or RBC glycerol concentration. Findings described herein revealed no effect of the intervention on aquaporine-3 content in general but showed higher glycerol levels within RBC after transfusion. Prior to transfusion, glycerol is removed from the stored RBC units to avoid hemolysis in the further course. Supernatant glycerol levels should thus be <1% (*w*/*v*), which was also the case in this study [57], and glycerol levels detected after transfusion were still below this reference value. The observed change might be attributed to transfusion, but the interpretation of the data should take into account that other factors including natural occurrence during lipolysis [58], adipose tissue [59] but also fasting [60] might affect blood glycerol levels. Finally, Na^+^/K^+^-ATPase was investigated because this enzyme determines intracellular ionic homeostasis and thus RBC volume described to affect RBC deformability [61]. The present study could not detect an effect of blood donation or transfusion on Na^+^/K^+^-ATPase content, although electrolyte concentrations were altered by this treatment as previously reported. Sodium levels increased during cryopreservation, while both potassium and calcium concentrations decreased [8]. Values measured herein support these recent reports because of reductions in potassium and calcium levels after transfusion. RBC electrolyte concentration show high inter-individual but also inter-day variability [62], and thus, further investigation is required to evaluate these changes.

Several study limitations need to be addressed. The small sample size, the fact that only male subjects were investigated, and the fact that elite athletes were not enrolled in this study might be regarded as study limitations. However, as ABD is prohibited, it was not possible to include active top athletes. In addition, monthly variations of female hormone concentrations and the use of hormonal contraceptives affect the investigation parameters, and thus might complicate the investigations of this cohort [63]. Yet, additional investigations are needed with larger and more diverse study groups to further investigate and explain the described changes. 

## 5. Conclusions

The results of the present investigation revealed a remarkable change of hematological parameters in response to RBC transfusion. However, those values remained within reference values for men. As stated by others, monitoring of hematological values related to ABD seems challenging and requires continuous measurements. The data additionally revealed that RBC age distribution shifted toward a younger circulating RBC population after the donation of large blood volumes. This shift distinctively increased RBC deformability in general and of the single sub-populations itself and might also be related to the drop in viscosity observed after the first blood donation. However, this requires further analysis. Transfusion did not affect viscosity or RBC aggregation. In contrast, parameters thought to affect RBC deformability including RBC membrane protein content were not altered during the study. Future studies should address activity analyses of the proteins because content and activity are not necessarily comparable. Finally, electrolyte levels seemed to be altered during the intervention, with potassium showing differences in the donation–transfusion phase. Related to an increase in RBC volume, maximum oxygen uptake was improved. Further, running speed at 4 mmol/L, as well as maximal running speed and running economy were improved after transfusion. The present study thus revealed that ABD significantly affects RBC deformability. However, the results need to be further evaluated in bigger and more diverse cohorts. Additionally, the underlying reasons for the changes in RBC deformability need to be further investigated.

## Figures and Tables

**Figure 1 biology-11-00647-f001:**
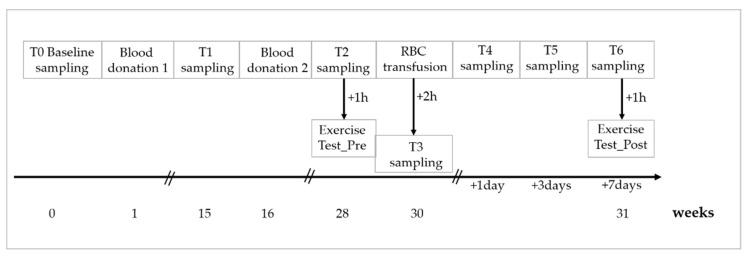
Schematic overview of the study design.

**Figure 2 biology-11-00647-f002:**
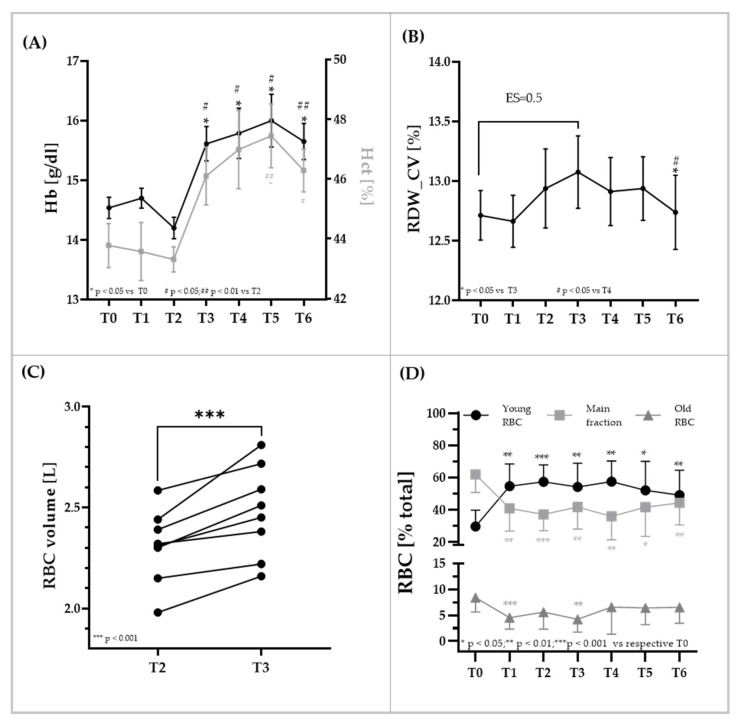
**Changes in RBC parameters during blood sampling and after transfusion.** (**A**) Hemoglobin concentration (black line) and hematocrit (grey line) significantly increased after RBC transfusion (=T3) and further increased until day three after transfusion (T5). Values decreased seven days after transfusion (T6). (**B**) Values of RBC distribution width might indicate an increase at time of transfusion (T3) and lower values observed seven days after transfusion (T6) compared to post transfusion (T3). (**C**) RBC volume significantly increased from T2 to T3. (**D**) RBC age distribution significantly changed after first blood donation. Percentage of young RBC (black) significantly increased after T0, while percentage of main fraction (light grey) and old RBC (dark grey) decreased, respectively. Data are mean (SD).

**Figure 3 biology-11-00647-f003:**
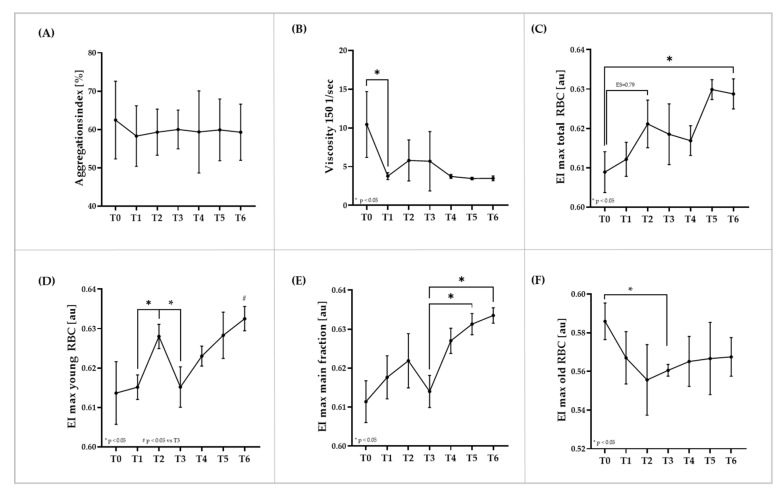
**Changes of rheological parameters due to blood sampling and transfusion**. (**A**) The aggregation index was not affected during the study period. (**B**) RBC viscosity decreased from T0 to T1 and remained at this reduced level throughout the remaining study period. Maximum deformability (EImax) of (**C**) total RBC increased throughout the study period, and of (**D**) young RBC increased during blood sampling (from T0 to T2) and from T3 after transfusion to T6. (**E**) EImax of main fraction RBC increased after blood transfusion. (**F**) In contrast, EImax of old RBC decreased throughout the blood sampling period (until T2) and remained reduced after transfusion. Data are mean (SD).

**Figure 4 biology-11-00647-f004:**
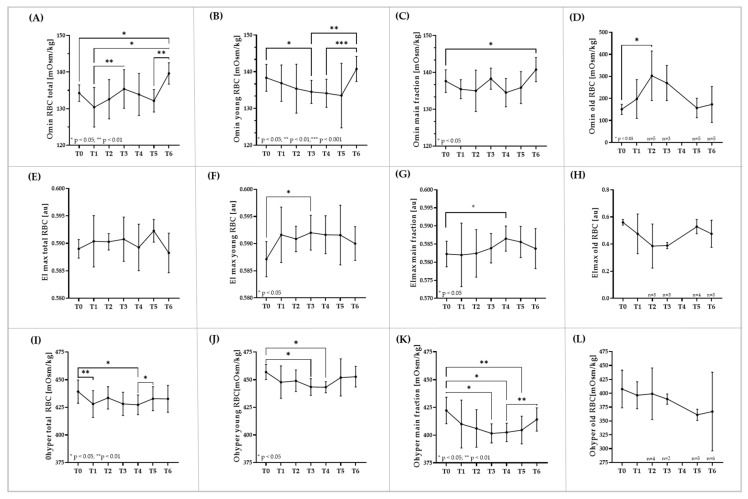
**RBC deformability under an osmotic gradient during blood sampling and transfusion period.** Omin (**A**) increased in the total RBC population during the study period, (**B**) while values decreased in young RBC, (**C**) increased in the main RBC fraction and (**D**) increased only during sampling period in old RBC. Maximum deformability EImax determined via osmoscan revealed (**E**) no changes in total RBC during blood sampling or post transfusion. Values increased in (**F**) young RBC and (**G**) main fraction and (**H**) reduced in old RBC during sampling period. Ohyper (**I**) of total RBC showed lower values during sampling while values increased during transfusion phase. Ohyper of (**J**) young RBC and (**K**) main fraction decreased during sampling but increased during transfusion phase. Ohyper of (**L**) old RBC decreased throughout the study period. Data are mean (SD).

**Figure 5 biology-11-00647-f005:**
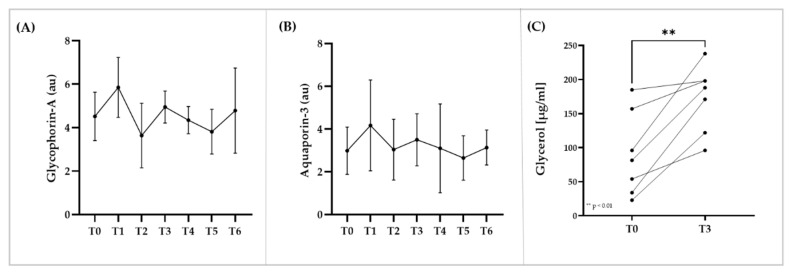
**RBC membrane proteins and changes in RBC glycerol concentrations during blood sampling and transfusion.** (**A**) Glycophorin-A and (**B**) Aquaporin-3 remained unaffected by the intervention. (**C**) RBC glycerol levels increased from T0 to T3. Data are mean (SD).

**Figure 6 biology-11-00647-f006:**
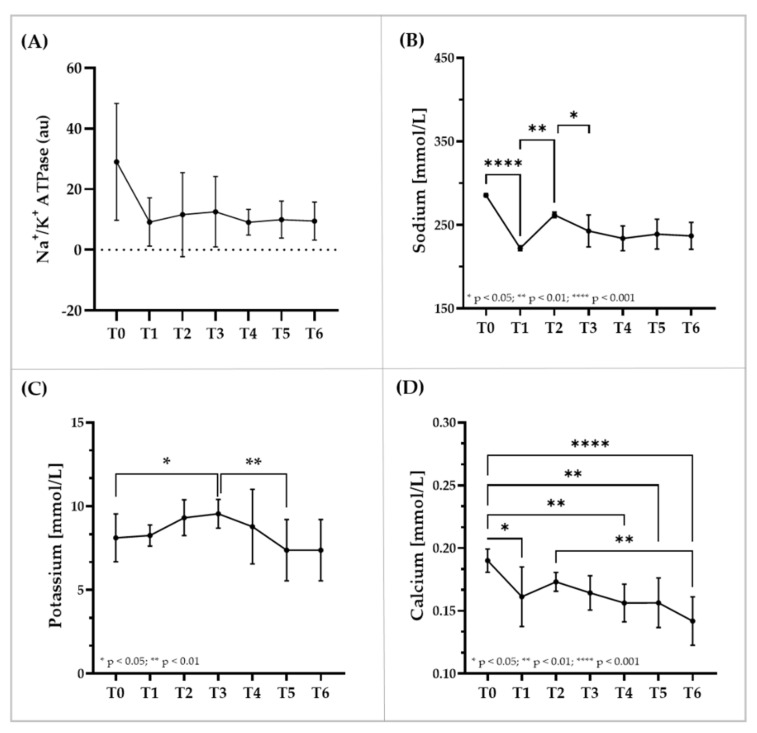
**RBC****Na^+^/K^+^**-**ATPase and changes in RBC electrolyte concentrations during blood sampling and transfusion.** (**A**) Staining of Na^+^/K^+^-ATPase remained unaffected by the intervention. (**B**) RBC sodium levels decreased after first blood sampling and after transfusion. (**C**) RBC potassium levels increased during sampling phase and decreased during transfusion phase. (**D**) RBC calcium levels decreased throughout the study period. Data are mean (SD).

**Figure 7 biology-11-00647-f007:**
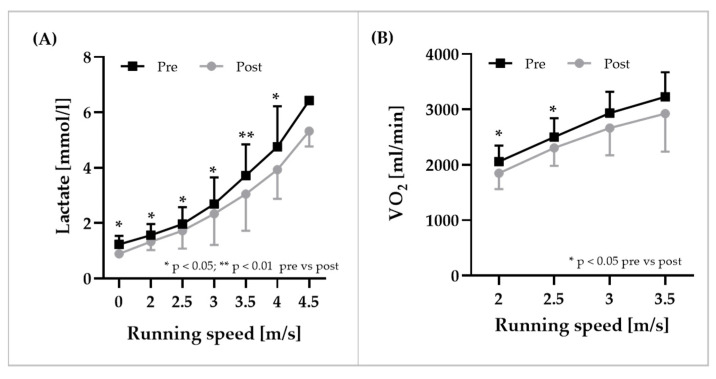
**Blood lactate concentrations and oxygen uptake during the submaximal incremental test.** (**A**) Blood lactate concentrations were significantly lower after blood transfusion, and (**B**) 60 s average oxygen uptake per increment decreased after blood transfusion. Data are mean ± SD.

## Data Availability

The data that support the findings of this study are available upon reasonable request from the corresponding author. The data are not publicly available due to privacy or ethical restrictions.

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
