# Peer review of "Autologous Blood Doping Induced Changes in Red Blood Cell Rheologic Parameters, RBC Age Distribution, and Performance"

_biology, 2022, doi:10.3390/biology11050647_

Round 1

Reviewer 1 Report

In this study by Dr Grau and coll. described changes in in red blood cell rheologic parameters, RBC age distribution, and performance in patients underwent autologous blood doping. It is also interesting to see how ABD is translated into indices of endurance performance, such as the maximal oxygen consumption, metabolic thresholds and exercise economy.

Despite the low number of cases included, analyses are well conducted and reported.

I have two minor comments:

  • To better interpret the data, I recommend to insert a table with some more detailed clinical and laboratory characteristics of the study participants.
  • I recommend English language revision.

Author Response

We would like to thank the reviewer for his/her time and effort to review our manuscript “Autologous blood doping induced changes in red blood cell rheologic parameters, RBC age distribution, and performance”. We would like to address the individual comments point-by-point.

Reviewer 1

In this study by Dr Grau and coll. described changes in in red blood cell rheologic parameters, RBC age distribution, and performance in patients underwent autologous blood doping. It is also interesting to see how ABD is translated into indices of endurance performance, such as the maximal oxygen consumption, metabolic thresholds and exercise economy.

Despite the low number of cases included, analyses are well conducted and reported.

I have two minor comments:

  • To better interpret the data, I recommend to insert a table with some more detailed clinical and laboratory characteristics of the study participants.
  • I recommend English language revision.

Response: We thank the reviewer for his/her evaluation. We included more details of the study participants in the section 2. Materials and Methods; Study group. We also revised language (spelling, grammar etc).

Reviewer 2 Report

In this study the authors investigate the dynamics of rheological (and associated) properties of healthy subjects undergoing autologous blood transfusion. While the overall data collection and analysis appears sound, without the contrast of an adequate control group, their insights are limited. Given this, I would suggest the authors appropriately limit how they frame their findings. Key comments below: 

Major
1) The authors make repeated comments about applicability to detecting of ABD. However, none of their analysis is in any way focused on this question. To study value of rheological parameters for detection of ABD the authors would need to compare parameters in subjects undergoing ABD to similar subjects without (for example, healthy subjects who donated blood and instead received a saline injection, to account for placebo effects). While the authors not some interesting temporal dynamics in their cohort, without a contrasting control group, they should not make any claims about relevance to detection of ABD. 

2) There are some points in the recruitment outline that need to be clarified. What population were the healthy subjects chosen from - local community, university, etc.? How did recruitment occur (flyers, emails, etc.). They exclude subjects based on "Nicotine-, alcohol-, or drug misuse" without defining what that is, and similarly for severe illness. They should also state the full list of potential confounding supplements that they excluded. 

3) A few of the non-clinically standard measurements (estimation of blood volume from Nadler equation, estimating RBC age from density measurements) are quite noisy (in associated literature). The authors should clearly discuss this limitation. 

4) At one point the authors suggest that exhibited results are likely insignificant purely due to sample size. They should remove this statement, as its misleading. 

Minor
1) Figure 1 is valuable, but needs to be made slightly clearer. For example, is the exercise test just pre- or post- the T2 blood draw? Given the short time period for T2-T5, more precise timings should be annotated on the figure. 

2) For Fig 2, it would be useful to plot the average change relative to the subject's starting point (e.g. % change in Hgb, etc.)

3) The authors should comment on why they see no significant change in RBC age groups after T3, as a change would physiologically be expected (as the transfused blood should contain older cells than the circulating blood, due to the increase blood production in response to the initial blood donation. 

4) Do the authors have any insight into why 2 subjects did not exhibit a large RBC volume change post transfusion, comparative to the others?

Author Response

We would like to thank the reviewer for his/her time and effort to review our manuscript “Autologous blood doping induced changes in red blood cell rheologic parameters, RBC age distribution, and performance”. We would like to address the individual comments point-by-point.

Reviewer 2

In this study the authors investigate the dynamics of rheological (and associated) properties of healthy subjects undergoing autologous blood transfusion. While the overall data collection and analysis appears sound, without the contrast of an adequate control group, their insights are limited. Given this, I would suggest the authors appropriately limit how they frame their findings. Key comments below: 

Major
1) The authors make repeated comments about applicability to detecting of ABD. However, none of their analysis is in any way focused on this question. To study value of rheological parameters for detection of ABD the authors would need to compare parameters in subjects undergoing ABD to similar subjects without (for example, healthy subjects who donated blood and instead received a saline injection, to account for placebo effects). While the authors not some interesting temporal dynamics in their cohort, without a contrasting control group, they should not make any claims about relevance to detection of ABD. 

Response: We agree with the reviewer that the interpretation of the data in terms of their practical application should be described more carefully. We have addressed this issue throughout the text.

2) There are some points in the recruitment outline that need to be clarified. What population were the healthy subjects chosen from - local community, university, etc.? How did recruitment occur (flyers, emails, etc.). They exclude subjects based on "Nicotine-, alcohol-, or drug misuse" without defining what that is, and similarly for severe illness. They should also state the full list of potential confounding supplements that they excluded. 

Response: We thank the reviewer for this point. We have included more details of our study population in the Material and Method section (sub-section Study population).

3) A few of the non-clinically standard measurements (estimation of blood volume from Nadler equation, estimating RBC age from density measurements) are quite noisy (in associated literature). The authors should clearly discuss this limitation. 

Response: We agree with the reviewer that the limits of these methods need to be addressed. We included these limitations within the text.

4) At one point the authors suggest that exhibited results are likely insignificant purely due to sample size. They should remove this statement, as its misleading. 

Response: Thank you for this critique. We deleted the sentence.

Minor
1) Figure 1 is valuable, but needs to be made slightly clearer. For example, is the exercise test just pre- or post- the T2 blood draw? Given the short time period for T2-T5, more precise timings should be annotated on the figure. 

Response: Figure 1 was revised according to the reviewers´ suggestion.

2) For Fig 2, it would be useful to plot the average change relative to the subject's starting point (e.g. % change in Hgb, etc.)

Response: We understand the intention of the reviewer to change the data presentation. However, we would like to keep the figure as it is to provide information of the actual values in order to allow the reader and experts in the field to actually see and understand the changes of the real values and to understand the interpretation of the data. Nevertheless, we included % changes between distinctive time points in the text and hope that the reviewer understands our argumentation.

3) The authors should comment on why they see no significant change in RBC age groups after T3, as a change would physiologically be expected (as the transfused blood should contain older cells than the circulating blood, due to the increase blood production in response to the initial blood donation. 

Response: The reviewer is right in that the transfused RBC unit should contain less young RBC compared to the high amount observed in the participants after the two blood donations. Calculation revealed that the transfusion resulted in a 3% decrease in young RBC, a 4.6% increase in RBC main fraction and a 1.3% decrease in old RBC (values at T3 compared to T2) indicating a slight shift. During cryopreservation, RBC ageing is reduced (see Bizjak et al. Cryobiology, 2018) and the thawing and washing procedure would eliminate fragile (and thus possibly old) RBC which would thus not enter the circulation. It is thus indicated that the transfused RBC also show a reduced number of old RBC, although this hypothesis needs to be addressed in further studies. Still, it was not expected that the chosen transfusion volume is able to highly affect the total RBC composition.  

4) Do the authors have any insight into why 2 subjects did not exhibit a large RBC volume change post transfusion, comparative to the others?

Response: We thank the reviewer for this critical point. We have critically evaluated our data and it seems that the values used in the first draft were calculated from values measured by the German Red Cross right before and immediately after the transfusion. So, these data did not reflect the time points T2 (two weeks before transfusion) and T3 (2h after transfusion). We now calculated the values from T2 and T3 and present those data within the manuscript. The differences might be explained by not yet complete equilibration of hematological values immediately after transfusion and might thus be a methodological problem. We believe the data presented now (really taken T2 and T3) are thus more clear.

Reviewer 3 Report

M. Grau and colleagues presented a fascinating study, the results of which will undoubtedly be of interest to both specialists in sports medicine and a wide range of researchers.
However, I have several comments for the authors.

1. In the manuscript, the authors point to a significant increase in RBCs deformability due to transfusion. Figure 3C shows that the EI max (total) increases after transfusion from 0.61 to 0.63. However, it remains unclear to what extent this change has clinical significance.

2. The authors of the manuscript do not provide characteristics of the RBCs transfused to subjects.

#. The authors do not analyze the correlation between the measured features (RBC deformability, young cells fraction, et. c) of subjects' blood and the markers (lactate and Vo2) of their physical condition.

Author Response

We would like to thank the reviewer for his/her time and effort to review our manuscript “Autologous blood doping induced changes in red blood cell rheologic parameters, RBC age distribution, and performance”. We would like to address the individual comments point-by-point.

Reviewer 3

  1. Grau and colleagues presented a fascinating study, the results of which will undoubtedly be of interest to both specialists in sports medicine and a wide range of researchers.
    However, I have several comments for the authors.
  2. In the manuscript, the authors point to a significant increase in RBCs deformability due to transfusion. Figure 3C shows that the EI max (total) increases after transfusion from 0.61 to 0.63. However, it remains unclear to what extent this change has clinical significance.

Response: The deformability of RBC (total RBC) increased throughout the intervention from 0.609 at T0 to 0.630 at T6 which resulted in statistical difference. The transfusion effect was not described for RBC in total but we did observe a difference after T3 for the young RBC. We agree that it remains speculative whether this would affect the blood flow in vivo.  

  1. The authors of the manuscript do not provide characteristics of the RBCs transfused to subjects.

Response: We agree with the reviewer that this would have been interesting information to complement our discussion. Unfortunately, it was not possible to sample RBC from the RBC bags prior to infusion because of safety issues and there were not enough RBC left in the blood bag after the transfusion for further investigations. But definitely, we have to address this point in future studies.

#. The authors do not analyze the correlation between the measured features (RBC deformability, young cells fraction, et. c) of subjects' blood and the markers (lactate and Vo2) of their physical condition.

Response: We agree with the reviewer that correlation analyses might help to depict relations of the different parameters. We performed correlation analyses between the proportion of young RBC and RBC deformability of total RBC. We further analyzed the correlation between RBC volume and maximum oxygen uptake. Both analyses revealed a relation between the tested parameters. The results are described in the manuscript.

Round 2

Reviewer 2 Report

The authors have addressed my concerns. I would still recommend a significant proof read of the manuscript before final publication, as there are multiple grammatical issues. However, these issues do not drastically limit the interpretability of the key findings.